Intelligent real-life key-pixel image detection system for early Arabic sign language learners

Alamri Faten S. 1
Rehman Amjad arkhan@psu.edu.sa 2
Abdullahi Sunusi Bala 3
Saba Tanzila 2
1 Department of Mathematical Sciences, College of Science, Princess Nourah Bint Abdulrahman University , Riyadh , Saudi Arabia
2 Artificial Intelligence & Data Analytics Lab (AIDA) CCIS Prince Sultan University , Riyadh , Saudi Arabia
3 Department of Electronics and Telecommunication Engineering, Faculty of Engineering, King Mongkut’s University of Technology Thonburi , Bangkok , Thailand
Pires Ivan Miguel
Electronic publication date: 2024 Jun 14
Publication date: 2024
Volume: 10
Electronic Location ID: e2063
Received 2023 Dec 27; Accepted 2024 Apr 25
Copyright: ©2024 Alamri et al.
Copyright year: 2024
Copyright holder: Alamri et al.
License: This is an open access article distributed under the terms of the Creative Commons Attribution License, which permits unrestricted use, distribution, reproduction and adaptation in any medium and for any purpose provided that it is properly attributed. For attribution, the original author(s), title, publication source (PeerJ Computer Science) and either DOI or URL of the article must be cited.
License URL: https://creativecommons.org/licenses/by/4.0/

Keywords: Computer vision, End-to-end deep neural network, Early sign language learning, Sign language detection, Hand gestures, Pattern recognition, Natural language processing

Funding: Deanship of Scientific Research, Princess Nourah bint Abdulrahman University through the Program of Research Project 44-PRFA-P-122 This research project was funded by Deanship of Scientific Research, Princess Nourah bint Abdulrahman University through the Program of Research Project Funding After Publication, grant No. (44-PRFA-P-122). Princess Nourah bint Abdulrahman University had a role in the study design, conduct, data analysis and interpretation (determined the scope of the work, directed the analysis, gave feedback and dictated the desired output/results verification), manuscript writing, and dissemination of results.

==============================
Lack of an effective early sign language learning framework for a hard-of-hearing population can have traumatic consequences, causing social isolation and unfair treatment in workplaces. Alphabet and digit detection methods have been the basic framework for early sign language learning but are restricted by performance and accuracy, making it difficult to detect signs in real life. This article proposes an improved sign language detection method for early sign language learners based on the You Only Look Once version 8.0 (YOLOv8) algorithm, referred to as the intelligent sign language detection system (iSDS), which exploits the power of deep learning to detect sign language-distinct features. The iSDS method could overcome the false positive rates and improve the accuracy as well as the speed of sign language detection. The proposed iSDS framework for early sign language learners consists of three basic steps: (i) image pixel processing to extract features that are underrepresented in the frame, (ii) inter-dependence pixel-based feature extraction using YOLOv8, (iii) web-based signer independence validation. The proposed iSDS enables faster response times and reduces misinterpretation and inference delay time. The iSDS achieved state-of-the-art performance of over 97% for precision, recall, and F1-score with the best mAP of 87%. The proposed iSDS method has several potential applications, including continuous sign language detection systems and intelligent web-based sign recognition systems.

Introduction

The hard-of-hearing population is growing from four hundred million to approximately five hundred million people in 2022 and has increased considerable attention to developing a hard-of-hearing communication solution (Abdullahi & Chamnongthai, 2022a). The hard-of-hearing population has only sign language as their mean of communication such as Arabic sign language, American sign language, Greek sign language, and Hausa sign language, etc. Sign language is the movement of the hand and body to transcribe an action that can be understood by the native speaker (native speaker means a person with a hard of hearing) or a professional signer (an expert with knowledge of sign language). Sign languages differ from one nation to another. Because of this, a unified Arabic sign language (ArSL) is promulgated by the League of Arab States (Alyami, Luqman & Hammoudeh, 2023). The lack of professional ArSL signers within the society renders the communication and participation of 289,355 hard-of-hearing miserable in the Kingdom of Saudi Arabia (Alharthi & Alzahrani, 2023; El-Dakhs, Ozfidan & Ibrahim, 2023) as difficulty in early language acquisition, social isolation, etc. Addressing the difficulty in early sign language acquisition, an automatic sign language detection and recognition system is proposed to bridge the communication gap between hard-of-hearing and normal-hearing people. Sign language detection (SLR) is a pattern recognition task that automatically recognizes or detects a sign in a video or an image and maps it to its equivalent gesture.

SLR can be grouped into three broad transcription, finger-spelling, isolated, and continuous SLR. Finger-spelling is applicable for alphabet characters, construction of words, and digits (Abdullahi & Chamnongthai, 2022b). These signs are mostly applied during early language learning of both native speakers and professional speakers. Detecting these signs is done by a finger-spelling or isolated sign recognition system. Because of the benefits of early language acquisition during the study and demonstrating out-of-vocabulary transcription, in this article, we mainly focus on alphabet detection in the early language learning stage and the development of an automatic real-life Arabic sign language detection system (iSDS). Automatic SLR systems are designed based on pose-based and appearance-based schemes. Pose-based schemes utilize the skeletal coordinates of the hand and/or body (Abdullahi & Chamnongthai, 2022b). The skeletal coordinates may be extracted using computer vision sensors or pose approximation methods. While appearance-based schemes utilize the image or sequence of images (Podder et al., 2023). The images are captured using a digital camera to reflect real-life applications. The appearance-based schemes are employed in the development of early Arabic sign language-based systems. Appearance-based detection models of early Arabic sign language are often lightweight and feasible for real-time detection in resource-limited devices such as ubiquitous systems, tablets, online/web-based recognition, etc. The most common recognition methods are handcrafted-based (Abdullahi & Chamnongthai, 2023), which requires sufficient expertise from native or professional speakers. The lack of sufficient image features and the inter-pixel dependencies among the image features from the handcrafted-based methods may lead to misinterpretation.

Recently, deep learning networks have revolutionized Arabic sign language recognition and reduced sign misinterpretation. More especially, convolution neural networks (CNNs) which automate feature extraction and recognition, CNNs have the inherent ability to extract important features that are difficult to capture and recognize handcrafted. The existing studies on Arabic CNN-based SLR models rely on two-dimensional convolution neural networks (2DCNNs) or three-dimensional convolution neural networks (3DCNNs) to extract appearance-based features (Aly & Aly, 2020; Abdul et al., 2021; Sidig et al., 2021).

Research gap

The existing studies on Arabic CNN-based SLR models rely on two-dimensional convolution neural networks (2DCNNs) or three-dimensional convolution neural networks (3DCNNs) to extract appearance-based features (Aly & Aly, 2020; Abdul et al., 2021; Sidig et al., 2021). Although these studies provide acceptable recognition accuracy, they are often more computationally expensive which hinders their deployment to real-time and real-life applications. In addition, CNN-based systems struggle with large-labeled images during training, which can be arduous to acquire in the sign language domain. Most existing methods have proposed the concept of transfer learning (TL). TL is particularly useful for a small sample size to achieve certain tasks. The pre-trained model provides a better starting point and avoids the retraining of a large model from scratch (Berriche, Alqahtani & RekikR, 2024). The limitations of the works in Latif et al. (2019), Sayed (2022) can be demonstrated in Fig. 1. As shown in Fig. 1, the ArSL characters “Thal” share common characteristics of shapes with the character “Zay” in frames e and f, respectively. Whereas the ArSL character “Ta” consists of similar hand pixels with the ArSL character “Thaa” in frames g and h, which confuses the detection algorithms of sign misinterpretation. As shown in Fig. 2A, the ArSL character “Thaa” shares similar characteristics of hand position and shape with character “Taa” in frame b. While ArSL character “Thal” has a similar hand position to ArSL character “Ta” in frames c and d, respectively. It can be explored from the figures that the reason for sign misinterpretation is the difficulty of capturing the hand position in space and the axis of rotation of the hand in the image. These difficulties affect the spatial correlations in the image. In this article, we propose an intelligent Arabic sign language detection system focusing on key-pixel image processing and inference delay time known as iSDS.

Research questions

This paper aims to address the following research questions (Rs), and the answers are provided in the results and discussion section.

R1: What is the influence of utilizing shape and rotation annotation schemes on the detection of the ArSL similar characters?

R2: How effective are the key pixels of each localized image frame in controlling the inherent problem of the ArSL image viewpoint?

R3:: Are the automatic inter-dependence pixel-based YOLOv8 features distinct to real-life noisy image detection?

Figure 1 The misinterpretation of similar ArSL signs by some of the existing ArSLR methods on Sayed (2022) data set.

Figure 2 The misinterpretation of similar ArSL characters by some of the existing ArSLR methods on Latif et al. (2019) data set.

The recent deep learning-based TL models (Latif et al., 2019; Sayed, 2022) open a direction for detecting a signer’s hand within the alphabet images with good performance. The limitations of the works in Latif et al. (2019), Sayed (2022) can be demonstrated by the ArSL characters “Thal” share common characteristics of shapes with the character “Zay” in frames e and f, respectively. Whereas the ArSL character “Ta” consists of similar hand pixels with the ArSL character “Thaa” in frames g and h, which confuses the detection algorithms of sign misinterpretation. However, the ArSL character “Thaa” shares similar characteristics of hand position and shape with character “Taa” in frame b. While ArSL character “Thal” has a similar hand position to ArSL character “Ta” in frames c and d, respectively. Batnasan et al. (2022) utilized the features of the You Only Look Once version 5 (YOLOv5) model to address the sign misinterpretation. YOLO is a CNN-based image detection network variant that has shown remarkable performance in TL-based sign language image recognition tasks and is widely deployed in computer vision applications (Batnasan et al., 2022). However, the YOLOv5 may fight with detecting sign language images with noisy backgrounds and shapes. It can be explored that the reason for sign misinterpretation is the difficulty of capturing the hand position in space and the axis of rotation of the hand in the image. These difficulties affect the spatial correlations in the image. In this article, we propose an intelligent Arabic sign language detection system focusing on key-pixel image processing and inference delay time known as iSDS. The data set is annotated despite missing hands or fingers within the frame. The missing hands may lead to limited performance during model learning. We employ a strategy of selecting key pixels of all the hand images by locating the exact hand position and size leading to designing a new image frame. The new image frames are effectively augmented to increase the diversity and robustness of the hand images. The key pixels from the augmented hand images are converted into the format that the YOLOv8 network can accept. The augmented images are fed into the YOLOv8 network. Interdependence-based pixel features and spatial correlations are proposed from the YOLOv8 network. Because of the limitations of the previous YOLO variants and the spurious regressions, a new version of YOLOv8 is introduced in the literature (Roboflow, 2023b). The YOLOv8 version has shown significant improvements in accuracy and speed, making it a viable option for real-life image detection, categorization, and segmentation. The YOLOv8 can achieve high accuracy while retaining a small model size and being deployed on AI-edge hardware or in the cloud at a reasonable cost. The YOLOv8 features are obtained by regularization and dimension reduction schemes. The features can capture the spatial dependency of the nearby pixels. The key pixels are identified and extracted at the early stage before feeding into the input layer of YOLOv8, this can improve the YOLOv8 learning and reduce the image dimensionality. The proposed iSDS method has the inherent ability to extract important features that are difficult to recognize by the standard CNN and former YOLO versions. This is the first article to extend the YOLOv8 algorithm for Arabic sign language detection in early sign language learning. To sum up, the contributions of this work are the following:

(a) The article selects key pixels of each image frame by finding and choosing pixels with large contrast using the effective distance between fingertip and wrist pixel for binary pixel processing. Selecting key pixels will control the inherent problem of viewpoint.

(b) The proposed method sets effective image annotations, average confidence threshold, and augmentation schemes for robust modeling and achieving high mAP.

(c) The article proposes inter-dependence pixel-based features using YOLOv8 layers to learn sign-distinct features that may be difficult to harvest using standard convolution methods for automatic real-life Arabic sign language detection.

(d) The proposed method is realized through an experiment using a web-based artificial intelligence and computer vision-based environment called Roboflow (Roboflow, 2023a) to demonstrate the system’s economic, real-life performance, and implementation efficacy.

The sections of the manuscript are as follows: The literature review is detailed in ‘Literature review’. The proposed method is given in ‘Proposed method’. The experiments are performed in ‘Experimental settings’. ‘Results and discussions’ presents the overall results of the experiments. The conclusion is given in section ‘Conclusions’.

Literature Review

In this section, the existing Arabic sign language detection system can be sub-grouped into non-deep learning-based systems and deep learning-based systems. The existing ArSLRs are developed using either 2D and/or 3D-based information with non-deep learning-based methods (Sidig, Luqman & Mahmoud, 2019). The methods combined modified Fourier transform-based, local binary, and histogram of gradients-based features for enlightening transitions of the hidden Markov layers. These methods demonstrate good recognition capability for Arabic gestures at the expense of complex segmentation and iterative processes. Hisham & Hamouda (2017) and Hisham & Hamouda (2021) extracts 76 positions with the help of a 3D sensor, and dynamic time warping. The positions were classified using the Adaboost method. The handcrafted features systems were not robust (Alnahhas et al., 2020; Deriche, Aliyu & Mohandes, 2019; Kammoun et al., 2020; Bird, 2022). Alternatively, Bansal, Wadhawan & Goel (2022) optimize the histogram of gradient features using embedded particle swamp optimization. Tharwat, Ahmed & Bouallegue (2021) develop a background invariant Quranic letter image recognition method using K-nearest neighbor for 14 alphabetic letters that represent the first Quran surahs in the Quranic sign language (QSL) known as AArSLRS. The AArSLRS achieved an accuracy of 99.5%. The handcrafted features depend on the specific feature being used for the ArSL recognition.

Furthermore, recently deep learning-based architectures have been extended for feature extraction or feature recognition and/or combining both as a single recognition system. Deep learning-based networks have demonstrated strong ability in SLR frameworks (Borg & Camilleri, 2020; Aly & Aly, 2020). One popular type of deep learning is the convolution neural network (CNN) which demonstrated superiority over handcrafted-based SLR methods (Abdullahi et al., 2024). Kamruzzaman (2020) proposed a CNN-based Arabic letter recognition network, where the generated feature maps are considered as a 3D matrix for classification and the final scores are manipulated for a speech-based generation. Kumar et al. (2017), Chong & Lee (2018) provide a skeletal and visual appearance-based position of the signer’s action for 3D SLR. Mittal et al. (2019) develop a recognition network using sequential layers by extracting the sub-units of the gestures, leading to a performance of 89.5%.

Bencherif et al. (2021) utilize a 3D-CNN skeletal network for the development of a new Arabic database. The database is created to achieve a near-optimal online sign recognition trade-off and an accuracy of around 90% is achieved for the signer-independent mode. Parelli et al. (2020) utilize the Openpose algorithm during hand trajectory estimation and the proposed features are transformed into 3-D. Kumar et al. (2020) proposed color-coded maps of the human 3D poses, however, a good performance is achieved at the expense of the feature transformation. Alternatively, active image sensors are robust in capturing tiny hand shapes and positions (Kumar et al., 2018), leap motion controller (LMC) (Lupinetti et al., 2020) etc., that allow natural interaction to the signer, while providing a high data stream. The limitation of deep learning-based methods is the extraction of general features that provide large-scale modeling.

De Coster, Van Herreweghe & Dambre (2020) improves the feature dimensionality through a transformer-based attention mechanism for effective sign feature localization. Zhou, Tam & Lam (2022) extract key image frame using the bidirectional encoding representation for the visual appearance-based model. Abdullahi & Chamnongthai (2022a) utilized maximal information correlation with cumulative match characteristics for feature ranking and selection of 3D skeletal hand poses. Latif et al. (2019) created a new ArSL data set of 32 letters using a digital camera known as ArSL2018. The ArSL2018 consists of 54,000 images which are further developed in CNN networks for classification and an average recognition performance of 92% is achieved. Zakariah et al. (2022) re-scale the weight of EfficientNetB4 from the transfer learning of effective learning of ImageNet data set for faster classification of 54,000 ArSL letters. The method improves the learning of standard CNN (Latif et al., 2019), however, the performance is not actualized in real-time scenarios. Likewise, reviews the progress of deep learning networks on ArSL classification and interpretation performance. Alyami, Luqman & Hammoudeh (2023) proposed Arabic sign language sequential manual and non-manual features from the transformer network. The transformer is trained from augmented MediaPipe poses + 33 landmarks and returns an accuracy of 68.2% from user independence mode. Podder et al. (2023) proposed features from the face-hand region-based segmentation and SelfMLP-infused MobileNetV2-LSTM-SelfMLP. An overall accuracy of 88.57%. Balaha et al. (2023) design a twenty ArSL word data set which is developed into the hybridized CNN-RNN network for recognition, yet an accuracy of 98% is obtained on signer independent mode. Alsulaiman et al. (2023) construct one of the Saudi sign language (SSL) databases known as the KSU-SSL database consists of 145,035 gestures. The KSU-SSL utilized the 3CGCN network for detection with 97.25% accuracy for the training and test split. However, the KSU-SSL data set is not a public data set. AbdElghfar et al. (2023) selected the 14 ritual-Quranic-based alphabet from the available 32 letters (Latif et al., 2019). AbdElghfar et al. (2023) recognize movements of the dashed Quranic letters to help the hard-of-hearing learn their Islamic rituals. Aldhahri et al. (2023) develop a method using MobileNet to consider the maximum accuracy and resource constraint of the ArSLR system. The method achieves a performance of 94.46%. Besides, feature selection schemes were used to control undesired features for effective sign representation (Al-Hammadi et al., 2020). Therefore, it is shown that the features extracted by a CNN are learned useful for developing an effective SLR system, in contrast to hand-crafted features that are designed beforehand by human experts to extract a given set of chosen characteristics. In addition, most of the existing deep learning models were too complex to be implemented in real-life applications. The existing deep learning-based methods for ArSLR are summarized in Table 1.

Table 1 Progress of some of the existing ArSLR systems using deep learning-based methods.

			
Method	Uniqueness of the method	Performance (%)	
Latif et al. (2019)	54,000 ArSL data set of letters	92	
Kamruzzaman (2020)	CNN image feature map into text	90	
Abdul et al. (2021)	CNN-BLSTM-based features	89.82 and 93.10	
Zakariah et al. (2022)	scale EfficientNetB4	95	
Batnasan et al. (2022)	Features of YOLOv5l	98.96	
Podder et al. (2023)	MobileNetV2-LSTM-SelfMLP	88.57	
Alyami, Luqman & Hammoudeh (2023)	TCN for MediaPipe poses	SD 99.7 and SI 68.2	
Balaha et al. (2023)	Hybridized CNN-RNN	98	
Alsulaiman et al. (2023)	Edges and vertices of 3D-GCN	97.25	
AbdElghfar et al. (2023)	Augmented Quranic-based images	99.54 at 42 min	
Aldhahri et al. (2023)	MobileNet for ArSL letters	94.46	

Proposed method

The proposed real-life intelligent-based Arabic sign language detection for early sign language learning (iSDS) is detailed in the following sections. Firstly, the images are annotated and labeled. Secondly, images are augmented and annotated as well. The two different sets of images are combined and their bounding boxes are created with the aid of JSON library. We detail the key pixel selection strategy from over fourteen thousand images, where each image of the frame is selected according to the background estimate. The final processed key pixel images are converted to be compatible with the YOLOv8 network. The inter-pixel features are automatically obtained from the YOLOv8 layers. Finally, the obtained features are called the sign-specific features. We trained the YOLOv8 with sign-specific features. The overall framework of the proposed method is summarized under pipeline subheadings as illustrated in Fig. 3.

Figure 3 The pipeline of the proposed iSDS method.

ArSL image formatting and upload

The images from the ArSL2018 and ArSL21L databases are uploaded in their original formats without any preprocessing or conversion. However, the ArSL2018 images are in grayscale format. The ArSL images are uploaded into the Roboflow computer vision-based environment for annotation, and Python application programming interfaces (APIs).

ArSL image annotation

Since YOLOv8 has an easy annotation scheme, That is implemented from a modified version of the Darknet annotation format. Therefore, we follow the same scheme, in which each ArSL image sample has one .txt file. The file consists of a single line in each bounding box. As shown in Fig. 4 first digit indicates the class ID to map the class name, while the remaining digits are space delimited and normalized between zero and one to locate the ArSL image. Note that each field is space-delimited and the coordinates are normalized from zero to one.

Figure 4 The sample of annotation file.

ArSL image augmentation

Image augmentation is a popular deep-learning process employed in CV to add the image size and diversity of the ArSL image set. Augmentation is particularly needed for ArSL sign detection because the ArSL sign images are limited by the challenges of occlusion and viewpoint changes. Therefore, several sign language image augmentation schemes were applied. The iSDS method employs the following effective image augmentation strategy.

(a) Flipping of ArSL images: In the iSDS, the images are flipped vertically and horizontally to enhance the YOLOv8 learning to detect hand gestures from all phases of the image frame.

(b) Rotation of ArSL images: We rotate the ArSL images to augment the image viewpoint angle of hand gesture between −15° to +15°.

(c) Scaling of ArSL images: The reason for scaling the ArSL images is to aid the YOLOv8 in learning to detect Arabic hand gestures of various sizes.

(d) Cropping of ArSL images: The iSDS method cropped the ArSL images to imitate the influence of pixel occlusion, which could lead the YOLOv8 to detect ArSL signs when were incompletely obscured. The ArSL images are cropped into 0% minimum zoom and 15% maximum zoom, respectively.

(e) Blurring of ArSL images: The reason for blurring ArSL images is to assist the YOLOv8 in detecting ArSL signs that are captured with bad lighting qualities. The ArSL images are blurred up to 1.5 pixels.

(f) Color manipulation: The color characteristics of ArSL images are automatically adjusted to assist the YOLOv8 in detecting ArSL signs in various lighting situations. The ArSL images are stretched to 512 × 512.

Bounding box design properties

In the proposed iSDS method, the bounding box is implemented using the JSON libraries in Roboflow (2023a). The bounding box reduces the search for the hand features in the image frame. The ArSL image bounding boxes are drawn using the two distinct rules:

(a) The ArSL image center point must always consist of a and b coordinates.

(b) The ArSL image must have corner points of a1, b1, a2, and b2 that can be obtained from Eqs. (1)–(4) with w and h denotes the image width and height, respectively. (1) a1=a−w2

(2) b1=b−h2

(3) a2=a+w2

(4) b2=b+h2.

The Eqs. (1)–(4) are used to show the basics that render the boxes within the ArSL images. The formulation allows the designing of faces of the bounding box viz; left, top, width, and height. The Eqs. (1)–(4) are provided to demonstrate the basic image points that the bounding box captures for localization and identification. The proposed iSDS method automatically utilizes the position and size of the hand in an image as set in the Eqs. (1)–(4). Sample ArSL images after the complete implementation of the annotation and augmentation schemes of the proposed iSDS method are illustrated in Fig. 5.

Figure 5 The result of the ArSL image annotation and augmentation.

ArSL image pixel processing

In this section, image pixel processing is proposed invariant to different background noises in YOLOv8. The iSDS method develops an effective pixel processing scheme using Euclidean distance to control the problem of viewpoint (that is some pixels are outside the bounding box) for compatible and accurate real-time YOLOv8 detection. The image pixel processing is realized by first generating the image pixels across the wrist and fingertip joint, respectively. The pixels are generated as follows:

(a) The bounding boxes that accommodate all image coordinates are returned according to the formulation in Eqs. (1)–(4).

(b) The image pixels that fulfill the test α in Eq. (5) are utilized for the pixel matching in line 10. All pixel coordinates a, b that fulfill the test condition are generated in lines 9–13 of Algorithm 1 . The iSDS method generates pairs of neighboring pixel coordinates according to the edge of each gesture across the bounding box from lines 14–25 of Algorithm 1 . The new coordinates of the bounding box are calculated in the loop from the gesture coordinates as explained in lines 26–27. The pixels of non-consecutive areas that fulfill the threshold condition which were outside the bounding box and did not form any joint are obtained using the process in line 29. However, the complete pixels across the gesture bounding box are output as explained in lines 30–31. Therefore, pairs of neighboring pixels that fulfill the threshold and contrast are represented as the gesture boundary of the wrist to the fingertip, and every pixel is viewed as its neighbor according to the definition in lines 32–33. The contrasting pixels are the exact sequence of each gesture across the bounding box.

The pixel processing uses the Euclidean distance to compute the distance between a set of hand key pixels P as explained in Algorithm 1 . We set pixels Fpixels with large contrast and distinction which effectively replace and discriminate similar ArSL signs to improve the YOLOv8 model’s dependency. Therefore, the distance between the fingertip and the wrist pixel is considered a sign feature for binary pixel processing Γ, however, a distance greater than an already decided threshold α is encrypted as one and vice versa. The encrypted pixels F are counted Q if the condition in step 41 of Algorithm 1  is satisfied and a further increment is made. The ArSL image pixel processing scheme fpx is given in Eq. (5). (5) fpx=1,Pn−0≥α0,otherwise

where fpx and Pn−0 denote the distance between the nth pixel of the fingertip and hand wrist, and substitute the pixel for the corresponding hand image. In addition, the goal of selecting key pixels of each image frame is to normalize each sign to make it independent of position and size; since each signer always does a different gesture for the same sign each time attempted.

Moreover, the key pixel selection is first pulled from the bounding boxes of detected objects in which each pixel within the region is related to the detected object including the background noise. Instead of using complex segmentation to choose the region of interest that may lose vital pixels leading to misinterpretation, we propose the key pixel selection to decide on the pixels with large contrast.

Intelligent sign language detection model

The You Only Look Once version 8.0 (YOLOv8) model is adopted to detect sign language characters. YOLOv8 is the timely single-stage neural network object detection model that recently demonstrated state-of-the-art real-life object detection performance. YOLOv8 architecture consists of an improved backbone (C2f) using convolution, advanced head, and neck as provided by the adopted YOLOv8 architecture in Fig. 6. YOLOv8 provides hand detection in an image by predicting bounding box(es) amongst an anchor-free detection head. The benefits of YOLOv8 from its predecessors are the provision of a large feature map and an enhanced inference time and precision of convolution. YOLOv8 exploits feature pyramid networks (P1-P5) for detecting objects of various dimensions. The YOLOv8 is enriched with a soft API for accessible integration in user applications. The YOLOv8 adopted in this article uses the Darknet-53 as a backbone network. The YOLOv8 developers (Utralytics, 2023) provide an effective loss function to measure the performance of bounding box coordinates, bounding box scores, and detection class scores. The most effective loss functions for the YOLOv8 detection module are the varifocalLoss, Bboxloss, and v8DetectionLoss which aim to enhance the detection accuracy of the model by overcoming limitations of scale variance, and foreground-background imbalance. For details of the YOLOv8 configuration readers are referred to Utralytics (2023). However, the summary of the YOLOv8 architecture is provided in the following steps of Fig. 6.

Figure 6 The structural representation of YOLOv8 modified in Utralytics (2023).

1. Input: hand image of size 512 × 512  × 3.

2. Output: a list of bounding boxes, class labels, and confidence scores for each detected hand image

3. Backbone = CSPDarknet53 (): Consists of 53 convolution layers with skip connections and 5 spatial pyramid pooling (P1-P5). It extracts high-level features from the input hand image through cross-stage partial connections.

4. Head = YOLOv8Head (): a set of convolution (Conv), upsampling (U), and concatenation layers (C) that process the backbone features. It consists of three output branches (P1-P3, Detect), each with a different scale. It predicts the bounding box coordinates, class probabilities, and objectness scores for each grid cell.

5. Loss = YOLOv8Loss(): a new loss function that combines focal loss, IoU loss, and label smoothing. It measures the difference between the predicted outputs and the ground truth labels. It reduces the false positives and improves the localization accuracy.

6. Model = YOLOv8(backbone, head, loss): a combination of the backbone, head, and loss. It takes a hand image as input and returns the detection results as output. This model is trained on hat data sets and the weight is fine-tuned on ArSL images.

The overall training and testing of the YOLOv8 network is performed using the images in Eq. (6). Equation (6) provide a simple vector concatenation ⟶ of both the raw ArSL image and augmented ArSL images as follows. (6) Ppixels=⟶RawI,AugmentedI.

The proposed features in Eq. (6) are developed in the YOLOv8 network to generate the inter-dependence pixels across the detection head. The detection head comprises the configuration of 25 convolutions and fully connected layers. The YOLOv8 configuration is summarized in Table 2. The output of the detection head is the detection probability P ˆ for the detection of ArSL sign with encoding labels P as follows Eq. (7). (7) P ˆ=aζl⋅Ppixels+bl

where a, ζl, and bl denote the activation function, weight, and bias matrix of the DarkNet-53 layer. We have chosen box loss, class loss, and object for the loss function from the existing YOLOv8 framework (Utralytics, 2023). All the YOLOv8 network models are trained in an end-to-end fashion. We evaluate the performance of the proposed iSDS method on real-life Arabic sign language data sets.

Table 2 Setting of iSDS detection model.

Algorithm	Design	Options	
Image augmentation	Image flipping	Hor. & vert.	
Rotation	−15° and +15°	
Scaling	stretch to 416 × 416	
Cropping	0% min., 15% max. zoom	
Blurring	up to 1.5pixel	
Color manipulation	150	
YOLOv8	Adam	0.9	
Batch size	16	
IOU	0.5:0.95	
Epochs	250	
Learning rate	0.01	
Weight decay	0.001	
Drop out	0.5	
Layers	225	
	Parameters	8,036,976	
	Regularizer	l2	

Data sets for iSDS verification

The proposed iSDS method is experimentally verified using the following public real-life Arabic sign letters and characters.

Arabic sign language data set (ArSL2018). The ArSL2018 images were taken at Prince Mohammad Bin Fahd University by volunteers of different age groups. Volunteers were made to stand around one meter away from the standing camera. Variations of images were introduced with different lighting, angles, timings, different backgrounds, and RGB format. The total number of captured images per alphabet was 54,049 images. The images were taken with various sizes and backgrounds for research consumption. The collected images were resized to a fixed dimension 64 × 64 and converted to grayscale images, with a range of pixel values from 0 to 255 (Latif et al., 2019).

ArSL21L: Arabic sign language letter dataset. Is a collection of annotated Arabic sign language letters and characters (ArSL21L) consisting of 14,202 images of 32 letter signs with various backgrounds collected from 50 people (Batnasan et al., 2022). Moreover, the data set could provide a first stage for early sign language learners to learn alphabets and characters which are the building blocks in word and sentence construction. The evaluation results of the data set revealed that it has a superior performance over the model trained on ArSL2018 (Sayed, 2022).

            _______________________Algorithm 1 ArSL image pixel processing algorithm  1:  start  2:  set P ←− I {image pixels}  3:  set α {decided threshold}  4:  set Γ {binary coding}  5:  set w,h {width and height}  6:  set v {coordinates}  7:  set Fpixels {target pixels}  8:  output Fpx  9:  repeat 10:     def match (I,α): 11:     w,h = I.size 12:     for a in range (w): 13:     − → for b in range (h) 14:     − →− → if α(P[a,b]): 15:     − →− →− → output a,b 16:     def v = set(v) 17:     for a,b in (v): 18:     − → if a − 1,b − 1 in (v): 19:     − →− → output (a,b), (a − 1,b − 1) 20:     − → if (a,b − 1) in (v): 21:     − →− → output (a,b), (a,b − 1) 22:     − → if (a + 1,b − 1) in (v): 23:     − →− → output (a,b), (a + 1,b − 1) 24:     − → if (a − 1,b) in (v): 25:     − →− → output (a,b), (a − 1,b) 26:     − → output (a,b), (a,b) 27:     box a,b = zip(v) 28:     output min(a),min(b),max(a),max(b) 29:     def non-consecutive(I,α 30:     for each in Γ(match(I,α)): 31:     − → output boundingbox (each) 32:     def contrasting (P): 33:     − → r,g,b = P 34:     − → output r¡50,g¡50,b¡50 35:     read i ∈ P 36:     while i = 1 ⋅⋅⋅N 37:     − → detect Pi 38:     − → encrypt Γ(Pn−0) − → Q 39:     end while 40:     compare Pi in Eq. 5 41:     if Q ≥ α : 42:     − → yes F = F + 1 43:     − → compute F ≥ Pn−0 44:     − →− → else go to step 10 45:     else if F=0 46:     − → go to step 10 47:     end if 48:     return  Fpixels 49:  until Fpixels ≥ α 50:  return  Fpx 51:  end________________________________________________________________________________________

Experimental Settings

In this section, we provide the experimental implementation of the proposed iSDS method. The simulation procedures and settings are explained in detail in the following subsections.

Image pixel standardization

The pixels of the ArSL images are preprocessed by calculating the mean µ, standard deviation std of both the raw-augmented images I and the label. Each feature was pre-processed via the µand std to control the noise and outliers as follows (8) φ=I−μIstd

where µ denotes the pixel mean of raw-augmented images. The normalized features are utilized as input to the iSDS detection model.

End-to-end Intelligent arabic sign language detection system

The network consists of thirty-two ArSL characters, making 32 YOLOv8 models. Each model is initialized with the weight of 100 image-trained data sets in Roboflow and fine-tuned with the automatic 14,202 image features of the ArSL21 data set. The improved YOLOv8 network using the proposed features is known as the iSDS network. The ArSL images are re-sized into 512 × 512 for the height and width respectively. The number of anchors is determined to be three and fine-tuned with thirty-two classes of the ArSL characters. The ArSL21 images are partitioned into 64%, 18%, and 18% for the training, validation, and testing phase, respectively. The testing phase is further employed separately from the training validation phase. The minimum confidence threshold of the YOLOv8 is chosen to be 0.25 with a non-maximum suppression (NMS) of 0.45 to suppress the bounding box overlapping. The iSDS network parameter selection is described in Table 2. We determine the number of network parameters by the YOLOv8 learning. The loss function is computed from the difference between the distance of the original image features concerning the predicted image features by the box loss, class loss, and object loss, respectively. The intersection over union (IOU) threshold of 0.5 is chosen to explore the overlap between the two bounding boxes. The designed iSDS models are trained with an Adam optimizer with suitable options β1 = 0.9, β2 = 0.99, and ϵ = 10−7 with a learning rate of 1 × 10−3. LeakyReLU activation function is utilized for proper presentation of non-linearity in the DarkNet. The second iSDS model is developed using the noisy image features of 32 ArSL characters of 54,049 image samples (Latif et al., 2019). The ArSL2018 data set has high variance, which is controlled using the proposed image pixel processing scheme. The YOLOv8 consists of thirty-two ArSL gestures, making 32 YOLOv8 models. All the models have been verified with different state-of-the-art evaluation metrics on the 64-18-18 image partitions. Pycharm with Anaconda IDE environment has successfully simulated each model with the Python code. Testing simulations of each model are evaluated based on the new signer in real-time.

iSDS parameter settings

Parameters influence the accuracy, precision, recall, and F1-score performance of the neural network. The parameters are chosen according to the YOLOv8 architecture to choose the number and size of the convolution layers. The number of DarkNet convolution layers is chosen to be 225 with 16 convolutional filters. The second parameters are chosen according to the preprocessing algorithm to choose the suitable values of image annotation and augmentation schemes. The YOLOv8 network settings are chosen from effective learning rate, batch size, number of training epochs, dropout, and iteration. The following parameters in Table 2 are chosen for the iSDS development. The hort. and vert. denote flipping at horizontal and vertical, min. and max. denotes minimum and maximum zoom.

Performance evaluation metrics

The performance of the proposed iSDS method is evaluated using the standard accuracy metrics and based on the strength of the deep learning size. The iSDS method is further evaluated using the effective metrics for real-time applications focusing specifically on inference time and detection speed as follows.

Accuracy

The accuracy of the proposed iSDS method is computed from the confusion matrix. The confusion matrix is categorized according to the tp, tn, fp, and fn which is defined as true positive, true negative, false positive, and false negative, respectively. (9) Accuracy=tp+tntp+tn+fp+fn.

Intersection over union

The intersection over union (IoU) is defined as the intersection between the predicted and ground truth bounding boxes divided by their union and varies from 0 to 1. The high value indicates high overlapping between the predicted and ground truth bounding boxes B1 and B2, respectively. (10) IoU=B1∩B2B1∪B2.

Mean average precision

To compute mean average precision (mAP), firstly, the average precision (AP) of each class representing the area under the precision-recall curve is computed. However, APs could be computed at various IoU thresholds and the mean of AP is obtained from the thresholds. Once the AP per class (ArSL sign category) is obtained, we can measure the mAP by averaging the AP values over all classes. We denote APi the average precision of the ith class and N the number of classes. (11) mAP=1N∑i=1NAPi.

Precision

Precision (Pr) demonstrates the proportion of positive recognition that was correctly detected, which is given as (12) Pr=tptp+fp

Recall

Recall (Rec) demonstrates the ratio of positive signs correctly detected by the model, it is given as. (13) Rec=tptp+fn.

F-1 score

F-1 score (F1) demonstrates the harmonic mean of precision and recall. It can be obtained from (14) F1=2tp2tp+fp+fn.

Frame per second

Frame per second (FPS) is defined as a unit that defines how fast the sign detection of YOLOv8 processes the input images and recognizes the desired target. It consists of the number of frames that occur each second. The higher the FPS value is, the faster the recognition will be.

Implementation and deployment of iSDS models

The proposed iSDS models are developed in a Pycharm IDE environment using the Python code in two different phases. The first phase is the implementation of the iSDS framework using the pipeline package. The image processing schemes are coded in Pycharm using the Numpy and PyTorch packages. The prepared code and processed images are uploaded into the Roboflow environment. The second phase is the iSDS model development using the Roboflow 3.0 object detection (fast). Roboflow provides a seamless environment for managing, preprocessing, augmenting, and versioning computer vision images and videos (Roboflow, 2023a). The Roboflow environment decrease 50% of the coding, assures quality of automatic annotation, saves training time, and increases model reproducibility.

Table 3 Evaluation results of the proposed iSDS method on ArSL characters.

ArSL characters	Precision	Recall	mAP_0.5:.95		ArSL characters	Precision	Recall	mAP_0.5:.95	
Ain	1	0.99	0.91		Laam	0.99	0.99	0.89	
AL	1	1	0.92		Meem	0.99	0.97	0.86	
Aleff	1	0.99	0.92		Nun	0.99	1	0.9	
Bb	0.99	1	0.92		Ra	1	0.99	0.84	
Dal	0.99	0.99	0.89		Saad	1	0.98	0.84	
Dha	1	0.98	0.86		Seen	0.98	0.98	0.82	
Dhad	1	0.99	0.84		Sheen	1	0.97	0.87	
Fa	0.98	0.99	0.86		Ta	1	0.99	0.89	
Gaaf	0.98	0.97	0.85		Taa	0.99	1	0.87	
Ghain	1	0.98	0.84		Thaa	0.99	0.98	0.82	
Ha	0.97	0.95	0.84		Thal	0.99	1	0.86	
Haa	0.99	1	0.91		Toot	1	1	0.87	
Jeem	0.99	1	0.86		Waw	0.99	1	0.83	
Kaaf	0.99	0.97	0.86		Ya	0.99	1	0.87	
Khaa	0.98	1	0.89		Yaa	1	0.98	0.86	
La	1	1	0.91		Zay	0.98	0.97	0.81	
					Average	0.99	0.99	0.87	

Results and Discussions

In this section, the iSDS performance results are presented in Table 3. The results are presented according to the standard evaluation metrics. In the table, the individual scores of each character from the ArSL21L data set are provided using Pr, Rec, and mAP metrics. The proposed iSDS succeeded in achieving the best Pr scores of ≥ 98% except at the ArSL letter Ha that returns the lowest precision score of 97%. The iSDS method returns an overall precision score of 99%. The recall (Rec) metric is evaluated on the detection performance of the proposed iSDS and most of the characters achieve best scores of ≥ 97% except the ArSL character Gaaf which achieved the lowest recall score of 95%. The overall average recall score of 0.99 (99%) is returned by the proposed iSDS. We further computed the mean average precision (mAP50:95) at 50–95 of the iSDS detection and the best mAP score of 92% is achieved, while the lowest score of 81% mAP is detected at character Zay. An average of 0.87 mAP score is achieved. The plots of the average detection performance of the proposed iSDS method are illustrated in Fig. 7. The detection performance of the iSDS method on the test images is illustrated in Figs. 8–10. The results show that the iSDS method can accurately detect new ArSL image characters 98% true even with similar hand shapes. The training progress of the iSDS method according to the mAP at various epochs is shown in Fig. 11, however, the best mAP performance is achieved at 250 epochs.

Figure 7 The average detection loss of the proposed iSDS method on ArSL data set.

Figure 8 iSDS performance on ArSL character Dhaa.

Figure 9 iSDS performance on ArSL character Saad.

Figure 10 iSDS performance on ArSL character Laam.

Figure 11 Training progress of the iSDS according to the mAP.

Figure 12 iSDS performance on ArSL character Sheen.

Furthermore, the YOLOv8 model initialized from the weight of 100 well-trained images of the red hat is on a single scale with approximately forty-three million parameters of 165.2 billion floating-point per second (FLOPs).

Deployment of the iSDS model for real-life and real-time application

The iSDS model is deployed for real-time and real-life application of unseen ArSL characters. The model is deployed using the JSON interface which is one major benefit of the YOLOv8 network, the flexibility of the interface with various computer vision web databases and devices. The iSDS model is configured with the webcam in real-time to detect the performance of the non-professional signer. The test image input to the model is the hand video. The signer is trained to perform any of the 32 ArSL characters of his choice in front of the webcam as shown in Fig. 12. The detection performance of the deployed iSDS model is illustrated in Figs. 12–14. The figures show the deployment of the proposed iSDS pipeline on the computer vision-based online environment (known as Roboflow). We generate the ArSL detection API from the Roboflow using the JSON interface. The Roboflow environment enables us to see the performance of the iSDS on web-based real-life applications. The deployment of the iSDS model on Roboflow enables us to observe the signer independence mode of the proposed method and to track and visualize the evaluation metrics in real time during ArSL hand image testing. We employ and task non-native speakers across the webcam to transcribe any ArSL alphabet of their choice from different backgrounds. The testing process investigates the capability of the detector to detect new image pixel features. Specifically, iSDS attains an increase in an average accuracy rate of 2% from all new signs over the existing methods as presented in Fig. 14. The detection of 92% assures that each bounding box contains the expected ArSL character unbiasedly. The detection performance demonstrates that the improved hand ArSL images using the key pixel image processing in the YOLOv8 model provide a robust detection method.

For effective analysis of the proposed iSDS method, we provide the box detection performance. The loss function for bounding boxes considers the confidence of the predicted bounding box, the IOU of the predicted bounding box, and the ground truth bounding box as shown in Fig. 15. It shows that we achieved a loss of less than two percent across the 250 epochs. The class loss concerning the IOU and the ground truth of the new signer is illustrated in Fig. 16. It shows that an average class loss of ≤ 3 is achieved. Figure 17 shows the average detection loss of the images under test concerning the IOU and mAP, respectively. The iSDS method achieved an average of less than 2 image losses. Figure 17 demonstrates that the iSDS method is one of the best options for the real-life detection of early sign language characters for early learners.

Vector analysis is provided in Fig. 18 to visualize our developed ArSL images in the iSDS model according to their similarity and to explore where the proposed iSDS model is struggling. The dark brown circles show the image pixels that are poorly detected, whereas the pure white circles illustrate the precise detection of the image pixels. However, the percentage of white circles is higher than that of brown circles, demonstrating that the real-time and real-life application of the iSDS is acceptable.

Figure 13 iSDS performance on ArSL character ha.

Figure 14 iSDS performance on ArSL character Dal.

Figure 15 The average box loss of iSDS method.

Figure 16 The average class loss of iSDS method.

Figure 17 The average ArSL image loss of the iSDS method.

Figure 18 The vector analysis of iSDS model.

Performance comparison between the proposed iSDS method and some state-of-the-art methods on ArSL data sets

In this section, we present a comparison results between the proposed iSDS method and some state-of-the-art methods in ArSL sign detection. The comparison is made on the two publicly available ArSL data sets. The first ArSL data set is known as ArSL21L which consists of 32 ArSL characters. The proposed iSDS method is implemented on ArSL21L and the individual detection performance of each character is presented in Table 4. Table 4 provides the comparison according to the precision, recall, and mAP_50:95 metrics, respectively. The metrics were chosen according to the state-of-the-art method for fair comparisons. The overall evaluation results are presented in tabular forms with an average of accuracy, mAP, Precision, Recall, and F1-score in percentage index, respectively. The bold-faced columns and rows inside the tables show the best results and methods, respectively. Table 5 provides the overall comparison results of some best existing methods on the same ArSL2018 data set. Zakariah et al. (2022) achieves the best accuracy of 95% compared to the recent methods on the ArSL2018 dataset, however, the proposed iSDS method outperforms all the existing methods with an improvement of over 2%. Recently, Aldhahri et al. (2023) and Podder et al. (2023) utilized 47,697 samples from the ArSL2018 data set in the MobileNet and an average recognition accuracy of 94.46%, however, our proposed iSDS improves the accuracy of MobileNet with around 3%. In addition, our proposed iSDS method utilizes the complete 54,049 samples of the ArSL2018 data set, unlike the MobileNet.

Table 6 presents a comprehensive evaluation of the existing methods on the ArSL2018 data set. The comprehensive evaluation is performed according to the four mentioned metrics, however, in circumstances where the method did not report the evaluation metrics we have decided to use not reported (NR) in the column. The proposed iSDS and the Latif et al. (2019) achieve the best recall and F1 score. The proposed iSDS method improved the detection performance of the existing methods by around 3%. The overall precision, recall, and F1-score of the iSDS method are 98.43%, 97.28%, and 97.96%, respectively, which is superior to the existing methods as shown in Table 6. The quantitative result evaluation of the iSDS method is performed on the individual characters of the ArSL2018 data set and the results are compared with the best existing method Aldhahri et al. (2023) in Tables 7. 8 provides the overall comparison results between our proposed iSDS method with some best existing methods on the ArSL21L data set. The proposed iSDS method achieves the best accuracy of 98.10% which outperforms the methods in Sayed (2022). The proposed iSDS method improves the existing accuracy by about 6%. The comprehensive comparison of the ArSL21L data set is performed by evaluating the mAP, precision, recall, and F1-score in Table 9. The proposed iSDS method achieves the best mAP score of 0.87 compared with the existing score (Sayed, 2022) by 0.4. In a nutshell, it can be observed that the proposed iSDS method provides the best detection results on the two publicly available ArSL data sets for early sign language learning and confirms the state-of-the-art recognition performance in real-life. The proposed iSDS method answers the well-formulated research questions as follows.

Table 4 The quantitative evaluation of proposed iSDS method with SOTA method on ArSL21L data set.

Character	Batnasan et al. (2022)	Proposed iSDS		Character	Batnasan et al. (2022)	Proposed iSDS	
	ID	Pr	Rec	mAP_0.5:.95	Pr	Rec	mAP_0.5:.95			ID	Pr	Rec	mAP_0.5:.95	Pr	Rec	mAP_0.5:.95	
Ain	133	1	0.99	0.81	1	0.99	0.91		Laam	132	0.98	0.92	0.86	0.99	0.99	0.89	
AL	136	0.98	0.98	0.92	1	1	0.92		Meem	135	0.99	0.98	0.83	0.99	0.97	0.86	
Aleff	136	1	0.98	0.89	1	0.99	0.92		Nun	123	0.96	0.98	0.84	0.99	1	0.9	
Bb	137	0.99	0.99	0.89	0.99	1	0.92		Ra	123	0.99	0.99	0.78	1	0.99	0.84	
Dal	111	0.97	0.91	0.79	0.99	0.99	0.89		Saad	135	0.99	0.95	0.8	1	0.98	0.84	
Dha	135	0.98	0.98	0.84	1	0.98	0.86		Seen	135	0.98	0.98	0.87	0.98	0.98	0.82	
Dhad	130	0.98	0.98	0.83	1	0.99	0.84		Sheen	135	0.99	0.99	0.89	1	0.97	0.87	
Fa	135	0.94	0.96	0.82	0.98	0.99	0.86		Ta	135	0.97	0.97	0.85	1	0.99	0.89	
Gaaf	134	0.96	0.96	0.82	0.98	0.97	0.85		Taa	135	0.99	0.99	0.83	0.99	1	0.87	
Ghain	135	0.99	0.99	0.85	1	0.98	0.84		Thaa	137	0.99	0.99	0.85	0.99	0.98	0.82	
Ha	135	0.98	0.99	0.82	0.97	0.95	0.84		Thal	135	0.97	0.96	0.82	0.99	1	0.86	
Haa	135	0.97	0.96	0.72	0.99	1	0.91		Toot	135	1	0.99	0.84	1	1	0.87	
Jeem	135	0.97	0.97	0.77	0.99	1	0.86		Waw	121	0.96	0.98	0.79	0.99	1	0.83	
Kaaf	135	0.98	0.99	0.87	0.99	0.97	0.86		Ya	134	0.92	0.98	0.83	0.99	1	0.87	
Khaa	135	0.97	0.99	0.79	0.98	1	0.89		Yaa	135	0.98	0.99	0.82	1	0.98	0.86	
La	135	0.99	1	0.87	1	1	0.91		Zay	135	0.98	0.94	0.76	0.98	0.97	0.81	
										AVERAGE	0.98	0.98	0.83	0.99	0.99	0.87	

Table 5 Comparison of average accuracy between the proposed iSDS with some state-of-the-art methods on ArSL2018 data set.

ArSLR methods	Algorithms	No. of samples	Accuracy (%)	
Latif et al. (2019)	Standard CNN	54,049	92	
Bencherif et al. (2021)	3D-CNN	80 words	S.D 89.62 and 88.09 S.I.	
Batnasan et al. (2022)	ResNext2	54,049	45.47	
Bansal, Wadhawan & Goel (2022)	SVM	54,049	93.60	
Zakariah et al. (2022)	EfficientNetB4	54,049	95	
Aldhahri et al. (2023)	MobileNet	47,697	94.46	
Proposed iSDS	improved YOLOv8	54,049	97.38	

Table 6 Comparison of average mAP, Precision, Recall, and F1-score between the proposed iSDS with some state-of-the-art methods on ArSL2018 data set.

Methods	mAP (%)	Precision (%)	Recall (%)	F1-score (%)	
Latif et al. (2019)	NR	NR	NR	97.5	
Zakariah et al. (2022)	NR	95	95	95	
Aldhahri et al. (2023)	NR	93.15	92.46	92.99	
Proposed iSDS	94.80	98.43	97.28	97.96	

Table 7 Quantitative evaluation of proposed iSDS method with SOTA method on ArSL2018 data set.

Character	Aldhahri et al. (2023)	Proposed iSDS		Character	Aldhahri et al. (2023)	Proposed iSDS	
	Acc.	Pr	Rec	F1	Acc	Pr	Rec	F1			Acc.	Pr	Rec	F1	Acc.	Pr	Rec	F1	
Ain	1	0.99	1	1	0.99	0.99	0.98	0.99		Laam	0.98	0.99	0.98	0.98	1	0.98	1	1	
AL	0.96	0.98	0.96	0.97	0.98	1	0.98	0.98		Meem	0.99	0.95	0.99	0.97	0.96	1	0.96	0.94	
Aliff	0.99	0.99	0.99	0.99	1	1	1	1		Nun	0.99	0.99	0.99	0.99	1	0.98	1	0.98	
Ba	1	0.98	1	0.99	0.98	1	0.98	0.99		Ra	0.99	0.97	0.99	0.98	1	1	1	1	
Dal	0.98	0.99	0.98	0.99	0.98	0.98	0.97	0.97		Saad	1	1	1	1	1	0.99	1	0.98	
Dah	0.99	0.98	0.99	0.99	0.99	1	0.98	0.98		Seen	1	1	1	1	0.98	0.99	0.92	0.98	
Dad	0.99	0.99	0.99	0.99	0.99	0.98	1	0.98		Sheen	0.99	0.99	0.99	0.99	1	1	1	1	
Fa	1	1	1	1	0.99	0.95	0.97	0.96		Ta	0.92	0.98	0.92	0.95	0.96	0.96	0.97	0.94	
Gaf	0.97	0.96	0.97	0.97	0.98	0.92	0.98	0.93		Taa	1	0.99	1	1	0.98	0.97	1	0.99	
Ghayn	1	0.99	1	1	0.99	1	0.98	0.98		Thaa	1	0.95	1	0.98	1	1	0.97	0.98	
Ha	0.99	0.99	0.99	0.99	1	0.92	1	0.97		Thal	0.94	0.97	0.94	0.95	0.96	0.98	0.97	0.97	
Haa	0.94	0.98	0.94	0.96	0.96	0.98	0.96	0.98		Taah	1	0.96	1	0.98	0.98	1	1	0.99	
Jeem	0.96	0.99	0.96	0.97	0.98	1	1	0.99		Waw	1	1	1	1	0.94	0.97	0.93	0.98	
Kaff	0.97	0.99	0.97	0.98	0.98	1	0.99	0.99		Ya	0.99	0.98	0.99	0.98	0.96	0.98	0.98	0.97	
Khaa	0.99	0.99	0.99	0.99	1	0.99	1	0.98		Yaa	0.99	0.98	0.99	0.98	0.99	0.99	1	0.99	
La	0.95	1	0.95	0.98	0.98	1	0.98	0.99		Zay	0.99	0.99	0.99	0.99	1	1	1	1	

Table 8 Comparison of average accuracy between the proposed iSDS with some state-of-the-art methods on ArSL21L data set.

ArSL21L methods	Algorithms	No. of features	Accuracy (%)	
Batnasan et al. (2022)	YOLOv5l	14,202	95	
Batnasan et al. (2022)	ResNext2	14,202	91.04	
Proposed iSDS	improved YOLOv8	14,202	97.99	

Table 9 Comparison of average mAP, Precision, Recall, and F1-score between the proposed iSDS with some state-of-the-art methods on ArSL21L data set.

Methods	mAP (%)	Precision (%)	Recall (%)	F1-score (%)	
Batnasan et al. (2022)	99.09@0.5, 83.06@0.5:0.95	97.87	97.66	NR	
Proposed iSDS	99.80@0.5, 87@0.5:0.95	99.10	98.99	98.89	

R1

The implementation of the effective YOLOv8 annotation maps each class ID with its corresponding class name. The annotation is based on the orientation and shape of each hand allowing locating the exact ArSL image within the database. Utilizing the quality annotation with effective image augmentation schemes improves the detection performance of similar ArSL images. All training ArSL images are flipped vertically and horizontally. All training ArSL images are rotated between the viewpoint angle of −15° to +15°. All ArSL images are cropped into 0% minimum and 15% maximum zoom. All training ArSL images are blurred up to 1.5 pixels. The annotation scheme leads to automatic extraction and utilization of distinct hand shape and rotation features.

R2

The ArSL image key pixels are obtained by designing a bounding box for each ArSL image to ease YOLOv8 detection. The complete pixels across the gesture bounding box are considered the new image frame. Therefore, pairs of neighboring pixels according to the threshold and contrast are represented as the ArSL image boundary of the wrist to the fingertip. Every pixel is viewed as its neighbor. The contrasting pixels are the exact sequence of each ArSL image across the bounding box. The proposed new image frame (bounding box) leads to extracting the vital region of interest without complex segmentation. The new image frame allows to normalization of each sign image to make it independent of position and size. The utilized new image frame leads to attained maximum accuracy across different ArSL data sets with the best computational feasibility.

R3

The proposed iSDS method automatically extracts inter-dependence pixel features using YOLOv8. YOLOv8 provides a large image feature map that consists of all pixels within the bounding boxes. The pyramids (p1-p3) provide a feature map of different scales leading to control of the viewpoint problems that are overlooked by the key pixel bounding box. We achieve the best detection performance compared to the existing ArSL image detection networks.

Conclusions

We proposed an intelligent Arabic sign language detection system for early sign language learning in this work. The proposal of an effective image pixel processing scheme for the YOLOv8 network contributes both to effective image localization and detection feasibility thereby increasing the detection accuracy. The evaluation of the proposed iSDS method on two publicly available ArSL data sets shows that iSDS performed better than most of the state-of-the-art methods in the real-life detection of ArSL characters. Our results were shown to agree with and performed better than those of the existing best real-time ArSL method. The iSDS achieved a detection accuracy of 99% and 97% for ArSL2021L and ArSL2018 publicly available data sets. The detection results are achieved with not less than eighteen frames per second for each testing phase. Interestingly, the iSDS provides relatively low box loss and high mAP at 50–95% rating. Thus, optimization is unnecessary for the new signer as the iSDS framework on real-life deployment achieves more than 90% on unseen non-professional signers with 8 fps. The major limitation of the iSDS method is the grid search tuning of hyper-parameters which relies on a series of expensive computations, causing the problem of insufficient training efficiency. This also leads to a decrease in detection performance and model stability. In the future, we intend to extend the YOLOv8 detection to dynamic sign words by collecting a large database of annotated images and videos of various backgrounds and sizes. Finally, the contribution of the designed iSDS method can be embedded in systems for early sign language learning systems, AR sign gaming systems, and Quranic sign language applications.

Supplemental Information

Supplemental Information 1 Experimental Code

Additional Information and Declarations

Competing Interests

Author Contributions

Data Availability

The authors declare there are no competing interests.

Faten S. Alamri conceived and designed the experiments, performed the experiments, authored or reviewed drafts of the article, data curation, funding, and approved the final draft.

Amjad Rehman conceived and designed the experiments, performed the experiments, analyzed the data, performed the computation work, prepared figures and/or tables, data analysis, visualization, and approved the final draft.

Sunusi Bala Abdullahi conceived and designed the experiments, performed the experiments, analyzed the data, performed the computation work, authored or reviewed drafts of the article, writing original work and coding, and approved the final draft.

Tanzila Saba performed the experiments, analyzed the data, prepared figures and/or tables, review & Editing, and approved the final draft.

The following information was supplied regarding data availability:

The Arabic Alphabets Sign Language Dataset (ArASL) is available at Mendeley Data: Latif, Ghazanfar; Alghazo, Jaafar; Mohammad, Nazeeruddin; AlKhalaf, Roaa; AlKhalaf, Rawan (2018), “Arabic Alphabets Sign Language Dataset (ArASL)”, Mendeley Data, V1, doi: 10.17632/y7pckrw6z2.1.

The ArSL21L: Arabic Sign Language Letter Dataset is available at Mendeley Data: Gochoo, Munkhjargal (2022), “ArSL21L: Arabic Sign Language Letter Dataset”, Mendeley Data, V1, doi: 10.17632/f63xhm286w.1.

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
