# Peer review of "Intelligent real-life key-pixel image detection system for early Arabic sign language learners"

_PeerJ Computer Science, doi:10.7717/peerj-cs.2063_

## Round 0.1 · original submission · Major Revisions

Please revise according to the reviewers.

**Language Note:** The review process has identified that the English language must be improved. PeerJ can provide language editing services - please contact us at copyediting@peerj.com for pricing (be sure to provide your manuscript number and title). Alternatively, you should make your own arrangements to improve the language quality and provide details in your response letter. – PeerJ Staff

Reviewer 1 ·

Basic reporting

Here's the corrected version:

"I am very thankful to the authors for their valuable contribution in the field of sign language. This article proposes an improved sign language detection method for early sign language learners based on the You Only Look Once version 8.0 (YOLOv8) algorithm, referred to as the Intelligent Sign Language Detection System (iSDS), which exploits the power of deep learning to detect sign language-distinct features in real-time. The iSDS method could overcome false positive rates and improve accuracy as well as the speed of sign language detection.

The proposed iSDS framework for early sign language learners consists of three basic steps: (i) image pixel processing to extract features that are underrepresented in the frame, (ii) inter16 dependence pixel-based feature extraction using YOLOv8, and (iii) web-based signer independence validation. The proposed iSDS enables faster response times and reduces misinterpretation and inference delay time.

The iSDS achieved state-of-the-art performance of over 97% for precision, recall, and F1-score with the best mAP of 87%. The proposed iSDS method has several potential applications, including continuous sign language detection systems and intelligent web-based sign recognition systems.

Please see the following major revisions before the acceptance of this article:

- First, the start of the introduction is too poor. The author did not provide any details about the background and other significant details.
- References are not aligned in the complete article.
- Figure 1 should be discussed in the introduction and should be placed before contributions.
- Why did the authors divide the related work into different subsections?
- What is the need for section 3 in the place of the proposed work section?
- The need for equations 1, 2, 3, and 4 is completely missing in the manuscript.
- Experimental setup is well written.
- Results are well defined.
- There is a need for future work with explicit lines to cover a wider audience."

Experimental design

Included in basic reporting.

Validity of the findings

Included in basic reporting.

·

Basic reporting

## English language
Text needs careful editing. There are minor technical mistakes. The font used in the text is not the standard for the journal. Sometimes the text does not cover the requirements for academic style:
> different versions of YOLO (YOLOv2-v5) may have a fight with detecting sign language

## Intro, background and literature
I cannot agree with the following statement:
> You Only Look Once (YOLO) is a variant of CNN-based Arabic sign detection network
You Only Look Once networks model is not designed specifically for Arabic sign detection.

## Text structure

## Figures and tables
Table 1 goes outside the text boundaries.
Figure 6 is hard to read. It is too big, to detailed and the inscriptions are with a small font.

## Raw data
Within the text the authors claim that they propose:
> a real-time automated Arabic sign language detection system
However, the Python code shared by the authors is composed by a scrip in 81 lines and a script in 33 lines that work on static images, which does not correspond to the above claim.

Experimental design

## Originality of the research
The originality of the research is hard to be verified from the text.

## Definition of research questions
Research questions are not clearly defined. If the proposed work is focused on some image preprocessing for the aid the application of the neural network, it has to be clearly stated. If the work aims at modification of the YOLO model model (which cannot be verified by the provided source code) it has to be clearly explained how.

## Technical and ethical standards
The mathematical expressions used in the text has to be clarified and the notation has to be corrected.

Validity of the findings

## Impact and novelty
The impact and novelty are hard to be verified due to unclear aspect of the proposed research.

## Underlying data, statistically sound
The database of images used in the experiments are statistically meaningful. However, a system that processes them cannot be called *real-time*.

Additional comments

In my opinion it has to be clearly defined exactly what is done by the proposed method. The system that performs the recognition itself is the YOLO model, which is not proposed by the authors: it is an existing method.

---

## Round 0.2 · accepted · Accept

Based on the reviewer analysis, the manuscript can ve accepted. I personally revised the comments, and all comments were revised correctly.

Reviewer 1 ·

Basic reporting

The authors have carefully revised the article based on previous concerns. I recommend it for acceptance.

Experimental design

The experiments are well designed.

Validity of the findings

The results seems to be valid.

Additional comments

I suggest the acceptance of the article.